# Fecal Microbial Structure and Metabolic Profile in Post-Weaning Diarrheic Piglets

**DOI:** 10.3390/genes14061166

**Published:** 2023-05-26

**Authors:** Xianrui Zheng, Ke Nie, Yiliang Xu, Huibin Zhang, Fan Xie, Liming Xu, Zhiyong Zhang, Yueyun Ding, Zongjun Yin, Xiaodong Zhang

**Affiliations:** 1College of Animal Science and Technology, Anhui Agricultural University, Hefei 230036, China; 2Key Laboratory of Local Animal Genetic Resources Conservation and Bio-Breeding of Anhui Province, Hefei 230036, China

**Keywords:** weaned piglets, diarrhea, gut microbiota, fecal metabolites

## Abstract

(1) Background: Piglet diarrhea is one of the most serious diseases in pigs and has brought great economic losses to the pig industry. Alteration of the gut microbiota is an important factor in the etiology of piglet diarrhea. Therefore, this study aimed to analyze the differences in the gut microbial structures and fecal metabolic profile between post-weaning diarrhea and healthy Chinese Wannan Black pigs. (2) Methods: An integrated approach of 16S rRNA gene sequencing combined with LC/MS-based metabolomics was employed in this study. (3) Results: We found an increase in the relative abundance of the bacterial genus *Campylobacter* and a decrease in phylum Bacteroidetes and the species *Streptococcus gallolyticus* subsp. *macedonicus. (S. macedonicus)* in piglet diarrhea. Meanwhile, obvious changes in the fecal metabolic profile of diarrheic piglets were also detected, particularly higher levels of polyamines (spermine and spermidine). Moreover, there were substantial associations between the disturbed gut microbiota and the altered fecal metabolites, especially a strong positive relationship between spermidine and *Campylobacter*. (4) Conclusions: These observations may provide novel insights into potential etiologies related to post-weaning diarrhea and further enhance our understanding of the role of gut microbiota in host homeostasis and in modulating gut microbial structure.

## 1. Introduction

Diarrhea is the leading infectious disease that causes the retardant growth and death of post-weaning piglets and leads to great losses in swine production. Almost 49% of neonatal and young piglet deaths are caused by diarrhea [1]. Four mechanisms can lead to diarrhea, including osmotic diarrhea, exudative diarrhea, secretory diarrhea, and gastrointestinal hypermotility [2]. Piglet diarrhea is usually related to various factors, which mainly include genetic background [3] and environmental factors [4,5]. Recently, numerous studies referred to a decreased content in *Lactobacillus* bacteria and a loss of microbial diversity in post-weaning piglet diarrhea, whereas *Clostridium* spp., *Prevotella* spp., or Escherichia coli were positively impacted [6], which indicated that gut microbiota may be a leading cause in triggering diarrhea.

The gut microbiota of mammals offers many benefits to the host, including assisting digestion, production of vitamins and short-chain fatty acids, maintenance of normal functions in the intestinal mucosa, regulation of the host immune system, and protection against pathogens [7]. These have been reported from sterile animals, characterized by immature immune systems and imbalance of gastrointestinal functions, as a consequence of the absence of gut microbiota [8]. Withdrawal of sterile conditions or fecal transplantation could induce maturation of the immune system in germ-free animals [9]. An important factor in triggering several inflammatory bowel diseases such as Crohn’s disease and diarrhea is the aberrated structure and function of gut microbiota, which is also associated with the occurrence of opportunistic pathogens, such as *E. coli* and *Campylobacter* [4]. The loss of gut microbiota, within the mucus layer protecting the epithelium, during the weaning transition could make glycans more available for pathogenic microbiota [10], glycan derivatives coming from microbial degradation can promote the growth of pathogenic species. It has been suggested that the degradation of mucus polysaccharides by the commensal species Bacteroides thetaiotaomicron can release fucose, which can be used by pathogenic *E. coli* to activate type Ⅲ secretion system (T3SS) gene expression for recognizing and adhering to host enterocytes [11,12]. These observations highlighted the potential relationship of gut microbiota with post-weaning piglets’ diarrhea.

Meanwhile, the complex interaction of microbiota–microbiota and microbiota–host could release various metabolites, which compose the basic environment in the gastrointestinal tract and impact host health. Metabolites show the fastest response to environmental changes and play a major role in the response of a biological system to abiotic or other interference, thereby linking genotype to phenotype (which is produced by the flux of metabolites in a biological system) [13]. Sugiharto et al. [14] observed higher levels of piglet plasma proline, taurine, and carnitine after *E. coli* infection, while betaine, creatine, and L-arginine were lower compared to the control group. Wu et al. [15] demonstrated that diarrheal piglets showed higher concentrations of 4-aminobutyric and glycine in the jejunum caused by *E. coli*-induced diarrhea, which implied the potential role in ETEC infection. The close relationship of fecal metabolites with specific bacteria in patients with Crohn’s disease was suggested [16]. Further, a changed metabolic activity of gut microbiota in ulcerative colitis (UC) patients was also reported [17]. Recently, several studies found that butyrate, a major energy source of colonic epithelium and generated mainly from gut microbiota, could mediate differentiation of regulatory T cells by inducing transcription factor FOXP3 expression involved in host immune regulation [18]. Further research indicated activation of FOXP3 mainly due to histone H3 high acetylation on its promoter and enhancer under the action of butyrate [19]. Moreover, higher levels of butyrate mainly exert an anti-inflammation effect by regulating host immune tolerance to gut microbiota. While lower concentrations of butyrate mainly play a pro-inflammatory role, it could reconstruct the intestinal microbial structure by inhibiting the growth of pathogenic bacteria and accelerating the proliferation of butyrate-producing bacteria [20]. Extensive previous studies have helped us deepen our understanding of intestinal microbiota and their significant role in many biological processes. However, the role of gut microbiota in piglet diarrhea and the underlying mechanisms are still not clear. There have been no reports on the comparison of intestinal microbiota between healthy piglets and piglets with diarrhea in Wannan Black pigs.

In order to identify the key microbial groups that are responsible for the imbalance of intestinal microbiota in piglets with diarrhea, we utilized 16S rRNA gene sequencing to investigate the microbial structure of fecal samples from four diarrheic and four healthy weaned Wannan Black piglets. Additionally, we performed an LC/MS-based untargeted metabolomic analysis to identify potential metabolic biomarkers associated with post-weaning diarrhea. The results here may provide novel insights into developing new means for preventing or treating diarrhea in piglets.

## 2. Materials and Methods

### 2.1. Ethics Statement

All animal experimental protocols were carried out following the guidelines for the care and use of experimental animals formulated by the Institutional Animal Care and Use Committee of Anhui Agricultural University, Hefei, China, under permit No. AHAU 20190115.

### 2.2. Study Design and Animals

The piglets used in this study were from an experimental pig farm of the College of Animal Science and Technology, Anhui Agricultural University, and were maintained under identical husbandry practices. No antibiotic therapy was administered to any of the piglets. Previous history of viral infections was not found in the farm, such as PRRSV, PCV2, or diarrhea-related viruses. All piglets were exclusively sow-reared and were weaned at 25 d. Close watch was kept on the general health of all the piglets, with special attention given to fecal form and behavior. A piglet was considered to have diarrhea if their feces were watery or liquid for at least 5 days, while healthy piglets never experienced diarrhea or any other illnesses. The viscosity of feces was the defining factor in distinguishing between healthy and diarrhea piglets. Finally, a total of eight piglets (28 ± 3 days old, siblings or half-siblings, with a 1:1 male-to-female ratio) containing four diarrheic and four healthy with similar birth dates and parity were selected for the follow-up experiments.

The selected piglets were sacrificed after blood collection and used 75% alcohol for piglets, using sterilized sterile scalpel to open the piglets. Blood was collected in EDTA-containing and heparinized vacutainers after puncture of the jugular vein and plasma samples were harvested by centrifugation at 5000× *g* for 10 min at 4 °C. Plasma samples were stored at −80 °C until analysis. Meanwhile, feces were collected using sterile cotton swabs and immediately frozen in liquid nitrogen. Eight fecal samples (4 diarrheic and 4 healthy) were obtained for next analysis. The segments of jejunum and colon were removed and washed with ice-cold phosphate buffer saline (PBS), followed by fixing with paraformaldehyde.

### 2.3. Intestinal Morphology Detection

Two paraformaldehyde-fixed intestinal sections (jejunum and colon) were dehydrated and embedded in paraffin. Sections of 5 μm were cut and then stained with hematoxylin and eosin stain. Then, intestinal segments were observed using a light microscope with matched image analysis software.

### 2.4. Plasma Biochemical Parameters Analysis

*Diamine oxidase* (DAO) was used as a marker of intestinal injury. In this study, the content of DAO in plasma was detected using spectrophotometry based on the previous study [21]. The assay mixtures contained 3 mL phosphate buffer (0.2 M, pH 7.2), 0.1 mL (0.04%) horseradish peroxidase solution, 0.5 mL plasma, 0.1 mL o-dianisidine-methanol solution (0.5% o-dianisidine in methanol), and 0.1 mL of 75% cadaverine dihydrochloride. A total of 3.8 mL mixture was incubated for 30 min at 37 °C and absorbance was measured at 436 nm. The concentrations of endotoxin in the plasma were detected with a commercial kit (ToxinsensorTM Chromogenic LAL Endotoxin Assay Kit, Genscript, Nanjing, China) utilizing a modified limulus amebocyte lysate and a synthetic color-producing substrate to detect endotoxin chromogenically. The kit has a minimum endotoxin detection limitation of 0.01 EU/mL and a measurable concentration range of 0.01 to 1 EU/mL. The assay was performed in duplicate and according to the manufacturer’s recommendations.

The amount of plasma cortisol and noradrenaline was detected employing the competitive inhibition enzyme immunoassay technique with a functional sensitivity of 0.049 ng/mL for cortisol (Pig Cortisol ELISA Kit, CUSABIO, Wuhan, China) and 25.5 pg/mL for noradrenaline (ELISA Kit for Noradrenaline, Cloud-Clone Crop, Houston, TX, USA). The details were following the manufacturer’s instructions.

### 2.5. 16S rRNA Gene Sequencing and Sequencing Data Analysis

The 16S rRNA gene sequencing was performed to evaluate microbial structure including composition and diversity. Fecal samples were processed for bacterial genomic DNA extraction using TIANamp stool DNA kit (TIANGEN, Beijing, China). The quality and concentration of genomic DNA were estimated by a NanoDrop 2000 spectrophotometer (Thermo Fisher Scientific, Waltham, MA, USA) and agarose gel electrophoresis, respectively. The extracted DNA was used as a PCR template and the hypervariable V3-V4 region of the bacterial 16S rRNA gene was flanked by barcode primers. The primer sequences were 338F (5′-ACTCCTACGGGAGGCAGCAG-3′) and 806R (5′-GGACTACHVGGGTWTCTAAT-3′). Sequencing was performed on an Illumina HiSeq 2500 (Illumina, Inc., San Diego, CA, USA) following the manufacturer’s instruction for 2 × 250 bp paired-end reads.

The raw data were merged with FALSH (V1.2.7, Austin, TX, USA) [5] and quality control of sequences was performed using QIIME software [22]. Afterward, the sequences obtained were filtered with chimera [23] using UCHIME algorithm [24] by matching to the golden database. The high-quality sequences were clustered into distinct operational taxonomic units (OTUs) with a 97% similarity threshold by using UPARS software (v7.0.1001) [25]. They were then classified to distinct taxonomic levels (phylum, class, order, family, genus, and species) using RDP 3 classifier algorithm (V2.2) [26] by comparing sequences with SILVA132 database (SSU_Ref database v102) [27]. MicrobiomeAnalyst [28] was performed for analyzing α-diversity, which included calculations of observed species, Chao1, ACE, Shannon, and Simpson indices. ANOSIM analysis was performed using MicrobiomeAnalyst for β-diversity analysis based on Bray–Curtis distance. Linear discriminant analysis coupled with effect size (LEfSe) was used for identifying the bacterial taxa differentially represented between groups at genus or higher taxonomy levels based on Galaxy web application [29]. Finally, the software STAMP (v2.1.3) [30] was conducted to detect the differential bacterial taxa between two groups with a two-sided *t*-test, which was supplemented with LEfSe analysis. Statistical significance was considered with *P*-value less than 0.05.

### 2.6. Metabolites Extraction for UHPLC-MS Analysis

Homogenized fecal samples (100 mg) were resuspended with ice-cold water containing 80% methanol and 0.1% formic acid, vortexed, incubated on ice, and centrifuged at 15,000× *g* at 4 °C. The obtained supernatant was diluted and transferred to a fresh tube and centrifuged again. Finally, an appropriate supernatant was injected into an Hyperil Gold column (100 × 2.1 mm, 1.9 μm) for chromatographic analysis under the positive and negative polarity modes, respectively. The eluents were comprised of eluent A (positive mode: 0.1% FA in water. negative mode: 5 mM ammonium acetate, pH 9.0) and B (Methanol). The elution gradient program was: 2% B, 1.5 min, 2–100% B, 12.0 min, 100% B, 14.0 min, 100–2% B, 14.1 min, and 2% B, 17 min, the flow rate was 0.2 ml/min, column temperature was 40 °C. The Q-Exactive series mass spectrometer was operated in positive/negative polarity mode with a spray voltage of 3.2 kV, the capillary temperature of 320 °C, sheath gas flow rate of 35 arb and aux gas flow rate of 10 arb, and the data-dependent acquisition (DDA) procedure was used to MS/MS scan.

The raw data files generated by UHPLC-MS/MS were processed using Compound Discoverer 3.1 (Thermo Fisher, Waltham, MA, USA) to perform peak alignment, peak picking, and quantitation for each metabolite. The main parameters were set as follows: retention time tolerance, 0.2 min, actual mass tolerance, 5 ppm, signal intensity tolerance, 30%, signal/noise rate, 3, and minimum intensity, 100,000. The peaks were then matched with the mzCloud, mzVault, and MassList databases to gain accurate and relative quantitative results. All data were normalized by area normalization methods. The SIMCA 14.1 software (Umetrics, Umea, Sweden) and R (R version R-3.4.3) were used for multivariate variable pattern recognition analysis, including principal component analysis (PCA) and partial least squares discriminant analysis (PLS-DA). PCA was used to show the intern structure of the data and assessed stability of detector. PLS-DA was performed to obtain differences between groups and better explain the variables. The parameters R2X and Q2 were calculated to evaluate the explainable and predictive level of the model. A 200 times permutation was further conducted to check the robustness and predictive ability of PLS-DA model. Here, the intercept value of Q2 reflects the robustness of the model, the risk of overfitting, and the reliability of the model, the smaller the Q2 value, the better. Furthermore, the value of variable importance in projection (VIP) of the first component in PLS-DA analysis was obtained. The differential metabolites were filtered with variable importance in the projection (VIP) > 1, fold change (FC) > 1.2 or < 0.83, and *p* < 0.05.

The commercial databases including KEGG and MetaboAnalyst [31] were utilized to enrich the pathways of metabolites. Then, correlation analysis among plasma physiological parameters, differential microbiota (top 6 bacteria at genus and species level, respectively), and differential metabolites (key metabolites involved in enriched KEGG pathways) was performed to evaluate their potential functional interactions. The software CANOCO 5.0 [32] and TBtools [33] were used in RDA analysis and visualization of Spearman’s analysis results, respectively.

### 2.7. Statistical Analysis

Statistical comparisons were performed by two-tailed Student’s *t*-test or Mann–Whitney U test using IBM SPSS Statistics for Windows, version 26.0 (IBM Corp., Armonk, NY, USA). *p* < 0.05 indicated statistical significance.

## 3. Results

### 3.1. Differences in Morphological and Plasma Physiological between Diarrheic and Healthy Piglets

We compared the differences in intestine morphological structure between diarrheic and healthy piglets (Figure 1). Results showed that the jejunum and colon of diarrheic piglets (Figure 1B,D) exhibited abnormal morphology when compared to healthy piglets (Figure 1A,C). This was observed through abnormal histomorphological changes associated with intestinal mucosal injury, which included intestinal villus truncation, atrophy, and exfoliation.

In Figure 2, all measured hematological indices in piglets are presented. The findings indicated that the content of plasma DAO (Figure 2A), endotoxin (Figure 2B), cortisol (Figure 2C), and noradrenaline (Figure 2D) significantly increased (*p* < 0.01) in diarrheic piglets relative to healthy piglets.

### 3.2. 16S rRNA Gene Profile in Diarrheic and Healthy Piglets

To identify the diversity and composition of fecal bacterial communities in diarrheic and healthy piglets, we utilized gene sequencing on the V3-V4 hypervariable region of 16S rRNAA total of 8 fecal samples were collected, with an average of 63,751 effective tags obtained per sample (Appendix A). Taxonomic classification allowed for the identification of 1948 and 2067 operational taxonomic units (OTUs) in the fecal samples of healthy and diarrheic piglets, respectively, with a 97% sequence similarity threshold.

The relative abundance of the top 10 phylum and top 10 families from fecal bacteria between healthy and diarrheic piglets is shown in Figure 3A,B, respectively. *Firmicutes* were the most prevalent phylum in both healthy and diarrheic piglets, which accounted for 50% and 40% of the reads in healthy and diarrheic groups, respectively. They were followed by *Bacteroidetes*, *Fusobacteria*, *Proteobacteria*, and *Actinobacteria*, which accounted for 30%, 7%, 7%, and 2% in the feces of healthy piglets, respectively. The main bacterial composition in diarrheic piglets was broadly in line with healthy piglets, except for an unidentified phylum, which was the second most prevalent in the diarrheic group and accounted for 19% of the bacterial community. At the family level, *Streptococcaceae* (18%), *Prevotellaceae* (17%), *Ruminococcaceae* (11%), *Lachnospiraceae* (10%), Muribaculaceae (8%), and *Fusobacteriaceae* (7%) were the dominant bacteria in healthy piglets, but the content of *Campylobacteraceae* and *Peptostreptococcaceae* was lower than 1%. However, in diarrheic piglets, the latter two groups were the most dominant bacteria, accounting for 19% and 15% of total reads, respectively, and the relative abundance of *Muribaculaceae* decreased sharply from 8% to lower than 1%.

At the OTU level, OTU1 (*Streptococcus gallolyticus* subsp. *macedonicus)*, OTU18 (*Muribaculaceae*), and OTU4 (*Fusobacterium*) were the dominant microbiota in the feces of healthy piglets. OTU3 (*Campylobacter*), OTU4 (*Fusobacterium*), and OTU59 (*Alloprevotella*) were the major OTUs in diarrheic piglets (Appendix A).

### 3.3. Differences of Gut Microbial Diversity and Composition in Diarrheic and Healthy Piglets

The indices of ACE, Chao1, observed species, Shannon, and Simpson were calculated to estimate α-diversity, which showed insignificant differences between diarrheic and healthy piglets (i.e., healthy vs. diarrheic: 4.11 ± 0.44 vs. 3.30 ± 0.87 and 758.75 ± 167.70 vs. 680.60 ± 43.73 for Shannon index and observed species, respectively, Mann–Whitney U test, *p* > 0.05). The Venn diagram reflected the number of OTUs that were shared between groups as well as within groups, which showed there were 1621 common OTUs in two groups and 327 and 446 unique OTUs in healthy and diarrheic piglets, respectively (Figure 4A). Similarity analysis revealed a strong difference in microbial structure in feces between diarrheic and healthy piglets based on Bray–Curtis distance (ANOSIM, R = 0.341, *p* = 0.025 Figure 4B). 

Marked differences in bacterial composition between diarrheic and healthy piglets were further analyzed using LEfSe analysis. As shown in Figure 5, in healthy piglets, the bacterial community was predominantly from the phylum Bacteroidetes (including *Muribaculaceae*) and *S. macedonicus*, whereas in diarrheic piglets, *Campylobacter* was the most abundant. Combined with the *t*-test method, we found that the relative abundance of the phylum marked as unidentified Bacteria and the genus *Campylobacter* were significantly elevated in diarrheic piglets. The phylum Bacteroidetes, Gemmatimonadetes, the genus *Parabacteroides*, *Oribacterium*, unidentified *Prevotellaceae*, and other 16 genera exhibited decreased relative abundance in diarrheic piglets. At the species level, *S. macedonicus*, *Treponema* porciunm, Rumen bacteium NK4A214, and *Novosphingobium resinovorum (N. resinovorum)* were more abundant in healthy piglets, whereas *Acidobacteria* bacterium LWH4 and *Corynebacterium xerosis* were more abundant in diarrheic piglets (Appendix A). The results of *t*-tests were consistent with LEfSe analysis and *S. macedonicus* and *Campylobacter* may be key biomarkers in diarrheic piglets. This has a limited scope since all pigs (numbers are small) were from the same farm and were, in fact, siblings or half siblings. Therefore, these results have the most significance to this farm.

### 3.4. Metabolic Differences of Fecal Microbiota between Diarrheic and Healthy Piglets

Untargeted metabolomics analysis was used to assess the metabolic differences of fecal microbiota between diarrheic and healthy piglets. The PCA analysis showed obvious differences in the two groups, which indicated that the raw data obtained by UPHLC-MS technology were robust (Appendix A). A PLS-DA method was performed to better understand the different metabolic patterns. Figure 6A showed significant separation between two groups at the positive ions mode, the values for R2X, Q2, and the results of permutation tests indicated that the samples were of reasonable quality (Figure 6B). These results were consistent with the observations at the negative ion mode (Appendix A). A total of 312 differential metabolites were observed between diarrheic and healthy piglets at positive and negative ion mode, based on the standard with VIP > 1, FC > 1.2 or <0.83, and *p* < 0.05 (80 increased and 232 decreased) (Appendix A).

Comprehensive KEGG pathway enrichment analysis of differential metabolites between positive and negative ions mode showed arginine and proline metabolism, porphyrin and chlorophyll metabolism, β-alanine metabolism, cysteine and methionine metabolism, tryptophan metabolism, glutathione metabolism, glycine, serine, and threonine metabolism, and other pathways were altered (Appendix A). Further analysis, according to KEGG pathway diagrams, found that spermidine was the core differential metabolite, which can connect directly with arginine and proline metabolism, β-alanine metabolism, cysteine and methionine metabolism, glutathione metabolism, and glycine, serine, and threonine metabolism (Figure 7), indicating that spermidine could exert a key role in piglet diarrhea.

### 3.5. Correlation of the Fecal Microbiota, Plasma Physiological Parameters, and Fecal Metabolites in Diarrheic Piglets

We utilized Spearman’s correlation coefficient to construct correlation matrices which helped reveal possible functional relationships between the differential gut microbiota, the altered fecal metabolites (related to six enriched metabolic pathways), and the changed hematological parameters (including DAO, endotoxin, cortisol, and noradrenaline). As shown in Figure 8A, Campylobacter, the core differential bacteria in diarrheic piglets, was negatively correlated with *S*. *macedonicus*, *Treponema porcinum*, unidentified *Prevotellaceae* and *Lachnospira*, and *Campylobacter* were positively correlated with plasma physiological parameters (DAO, endotoxin, and noradrenaline). Whereas *S*. *macedonicus* and other differential microbiota in healthy piglets such as Rumen bacterium *NK4A214*, *Oribacterium,* and *Lachnospira* were negatively correlated with hematological parameters. For altered fecal metabolites in diarrheic piglets, spermidine was negatively correlated with indole and indole derivatives but was positively correlated with creatine, spermine, glycine, and all hematological indices (DAO, endotoxin, cortisol, and noradrenaline) (Figure 8B).

Finally, the correlation analysis was performed among differential microbiota (top six bacteria at genus and species level), changed metabolites (key metabolites involved in the enriched pathways), and the physiological state of piglets. The results of RDA analysis revealed that spermidine and xanthurenic acid were core metabolites with the conditional effect of 58.6% and 23.0%, respectively (Figure 9), and that spermidine, especially, could play a core role in microbial composition, these results supported those of KEGG pathway analysis (Figure 7). What is more, Campylobacter, *Corynebacterium xerosis*, and *Acidobacteria* bacterium LWH4 were positively correlated with spermidine and *S*. *macedonicus* and other bacteria were positively correlated with xanthurenic acid. In addition, the correlation of other altered metabolites with differential microbiota, based on Spearman’s method, was visualized in a detailed network diagram with the standard |r| > 0.5 and *p* < 0.05 (Appendix A), which is consistent with RDA analysis.

## 4. Discussion

In the present study, we used 16S rRNA gene sequencing and liquid chromatography–mass spectrometry-based metabolomics analysis to examine the impact of post-weaning diarrhea on the composition of gut microbiota and fecal metabolites. Our research revealed major differences in the microbiota and metabolic markers between diarrheic and healthy piglets. Specifically, we observed a decreased relative abundance of the species *S. macedonicus* and the phylum Bacteroidetes, including Bacteroides and Muribaculaceae, but an increased relative abundance of Campylobacter. Bacteroidetes are stable commensals in the gastrointestinal tract of mammals, including pigs, and they are among the early intestinal residents of normal piglets, most frequently found in the large intestine. By understanding the molecular functions of pivotal bacteria in the development of piglet diarrhea, our results may help develop strategies for preventing or treating this disease. These results have a limited scope since the piglets are from the same farm and the number of the piglets is small. Therefore, these results have the most significance to this farm.

The microbiota composition identified in the present study was consistent with previous studies, which demonstrated the results here were of high reliability. Previous study has shown that Firmicutes and Bacteroidetes were the two most abundant phyla in the gastrointestinal tract, followed by Fusobacteria, Proteobacteria, and Actinobacteria in mammals [34]. This study shows that the highest bacterial groups in the healthy gut, colon, and feces of piglets are Firmicutes, Bacteroidetes, Proteobacteria, Fusobacteria, and Actinobacteria, of which Firmicutes, Actinobacteria, and Proteobacteria are dominant bacterial phyla in the small intestine, accounting for more than 98% of the total proportion, while the dominant bacterial groups in the colon and feces are Firmicutes, Bacteroidetes, Fusobacteria, and Proteobacteria, accounting for an average of about 92%. The result is similar to previous findings in humans, mice, pigs, horses, chickens, and other animals, indicating that the composition of gut microbiota may have certain conservation among different species [34]. Thus, the dataset obtained in the present study may provide a valuable source for the pig microbiota database.

Important differences in intestinal microbiota composition between diarrheic and healthy piglets in the present study were revealed in the present study. Firstly, we found that the relative abundance of *Campylobacter* showed a significant increase in diarrheic piglets, which may be one of the major reasons for post-weaning piglet diarrhea. *Campylobacter*, a highly prevalent commensal bacteria in mammals, is recognized as the most common cause of acute food-borne bacterial diarrhea in humans, and *Campylobacter* jejuni and *Campylobacter* coli are responsible for ~85% and ~15% of cases, respectively [35,36]. Recent studies have found *Campylobacter* to be the sole pathogenic bacteria responsible for around 15% of diarrheic cases occurring in piglets [37]. Yang et al. [4] reported an elevated content of *Sutterella* and *Campylobacter* in pre-weaning piglets with diarrhea compared to healthy piglets. Li et al. [38] showed that *Campylobacterales* and *Campylobacter* increased after weaning in piglets and were one of the major reasons for post-weaning diarrhea. Thus, we assume that the moderately elevated abundance of *Proteobacteria* in fecal samples from diarrheic piglets might have accelerated the growth of *Campylobacter* to a certain extent. These findings will improve knowledge of the role of intestinal microbiota on piglet diarrhea and the underlying mechanisms.

We also observed a significant reduction in *Bacteroidetes and S. macedonicus* in diarrheic colonic samples compared to healthy piglets, which has been reported by Hermann-Bank et al. [2] and several other studies [39,40,41]. *Bacteroidetes* members are major proteolytic bacteria in the mammalian gastrointestinal tract and can produce butyrate. *Bacteroides* thetaiotaomicron has been observed to decompose the complex polysaccharide from fodder to aid in intestinal nutrient absorption [4]. Therefore, a significant reduction in *Bacteroidetes* can negatively impact normal gastrointestinal functions, such as digestion and absorption, and weakens repressive effects against pathogenic microbiota. The lactic acid bacteria *S. macedonicus*, has recently been discovered in the gastrointestinal tract of many animals including pigs, and is therefore most likely a part of the normal gut microbiota [42]. The S. *macedonicus* ST91KM strain has demonstrated the ability to produce a narrow-spectrum peptide that can combat pathogenic bacteria associated with mastitis in dairy cattle [43]. This study suggests that consistent reduction of *S. macedonicus* may hinder its ability to suppress pathogenic growth, potentially leading to piglet diarrhea. These results indicated that *S. macedonicus* strains show promise as fodder supplements acting as alternatives to antibiotics in the swine industry.

Beyond microbial composition, the microbial functional profiles in diarrheic piglets also differed from those in healthy piglets. In our current study, we found that elevated levels of spermidine appeared to be a core metabolite in piglet diarrhea and significantly impacted the microbial composition. It was reported that weaning stress caused alterations to the microbial metabolic profile in the intestine and resulted in variable levels of 433 metabolites, including amino acids, organic acids, and amines, potentially contributing to gut microbiota dysbiosis induced by weaning stress [38].

Previous studies have shown that polyamines play a crucial role in maintaining the differentiation and renewal ability of intestinal epithelial cells [44] and can promote the expression of specific genes, such as those for tight junction proteins [45]. Our research also revealed a strong positive correlation between *Campylobacter* and spermidine, suggesting a potential functional interaction between them. Hanfrey et al. [46] demonstrated that spermidine is critical for the growth of *Campylobacter* jejuni and is primarily synthesized through the aspartate β-semialdehyde pathway. Additionally, Nicholson et al. [47] suggested that spermidine plays an essential role in the toxic effects exerted by *Campylobacter* jejuni. Therefore, we can infer that spermidine-related metabolic pathways (arginine and proline metabolism, β-alanine metabolism, cysteine and methionine metabolism, glutathione metabolism, glycine, serine, and threonine metabolism) may be significant factors leading to post-weaning piglet diarrhea. Thus, the findings of the present study enhance our understanding of the role of pig gut microbiota in host homeostasis and could provide some theoretical reference for the prevention or treatment of post-weaning piglets’ diarrhea.

## 5. Conclusions

In conclusion, our study revealed differences in gut microbiota composition and metabolic changes between diarrhea and healthy piglets. *Campylobacter* was significantly increased, while Bacteroidetes and *S. macedonicus* were reduced in the diarrheic piglets. In addition, based on the metabolomics analysis, differential metabolites (i.e., spermidine) and metabolite-related metabolic pathways (i.e., arginine and proline metabolism) were identified as important pathways associated with diarrhea-induced gut microbiota dysbiosis. These findings may provide some theoretical reference for prevention or treatment of post-weaning piglets’ diarrhea. This study has some limitations, such as the small sample size and the fact that the piglets all came from the same farm. Therefore, the results of this study are more applicable to this specific farm.

## Figures and Tables

**Figure 1 genes-14-01166-f001:**
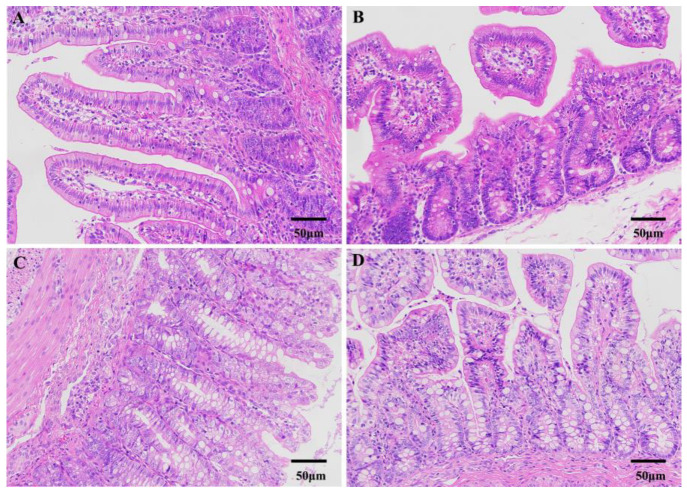
Intestinal morphological characterization in piglets at 50 μm level. Jejunum (**A**) and colon (**C**) of healthy piglets. Jejunum (**B**) and colon (**D**) of diarrheic piglets.

**Figure 2 genes-14-01166-f002:**
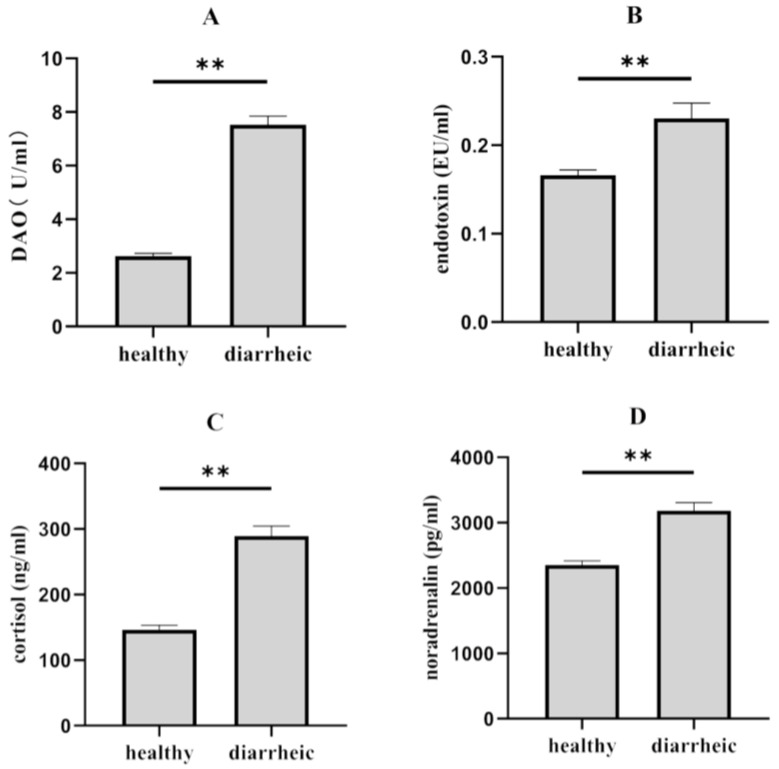
Changes in physiological parameters between healthy and diarrheic piglets in plasma. (**A**) DAO, (**B**) endotoxin, (**C**) cortisol, and (**D**) noradrenalin. Plasma biochemical parameters between diarrheic and healthy samples, which were visualized based on the means ± SEM. An independent *t* test was used to identify diferences between two groups. ** (*p* < 0.01) above the bars denote a significant difference.

**Figure 3 genes-14-01166-f003:**
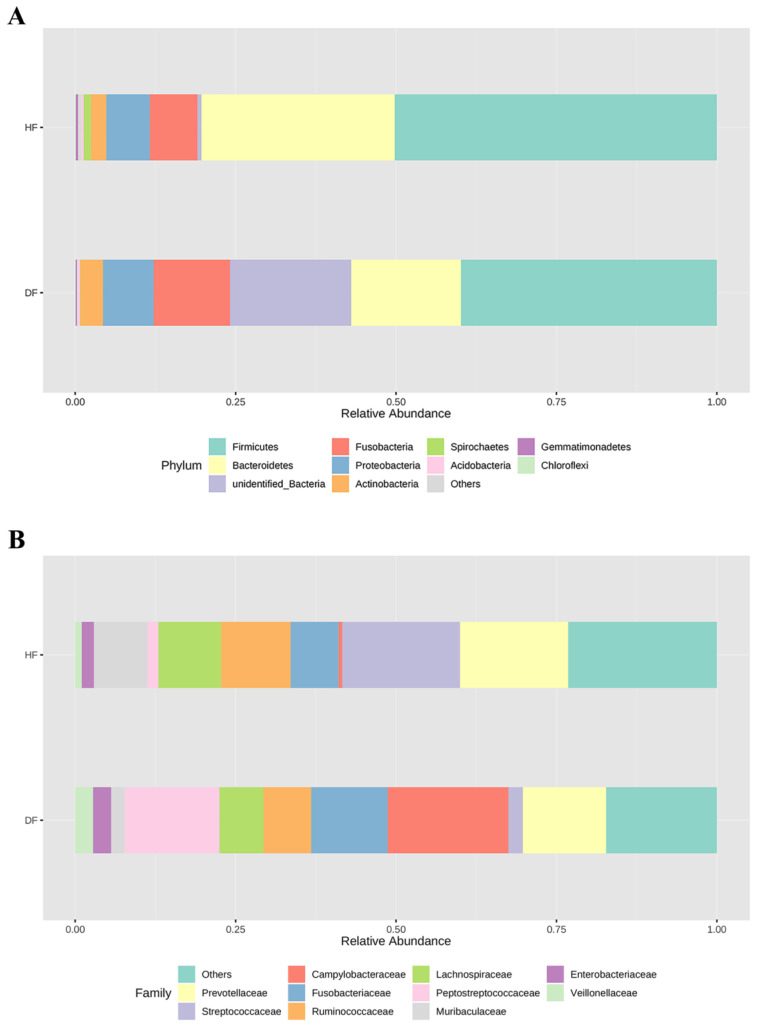
Gut microbial community structure in healthy and diarrheic piglets. The relative abundance of the top 10 phylum (**A**) and the top 10 families (**B**) of fecal microbiota in both healthy and diarrheic piglets. The fecal samples from healthy and diarrheic piglets are abbreviated as HF and DF, respectively.

**Figure 4 genes-14-01166-f004:**
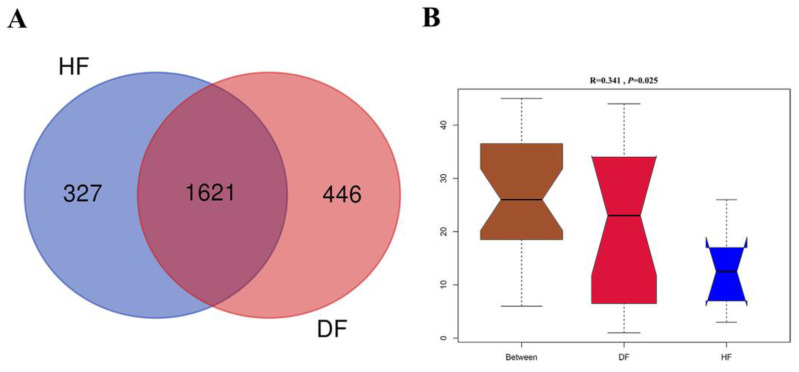
The overall structure of gut microbiota in healthy and diarrheic groups. (**A**) Venn diagram for bacterial OTUs compositions in two groups. (**B**) Similarity analysis (ANOSIM) based on Bray–Curtis distance.

**Figure 5 genes-14-01166-f005:**
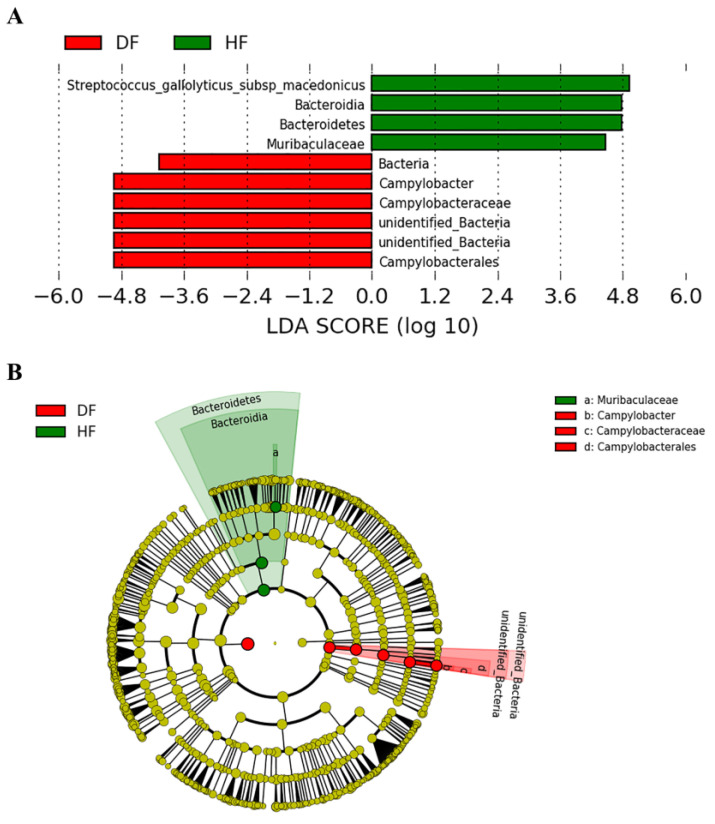
LEfSe analysis between healthy and diarrheic piglets. (**A**) LDA plot. (**B**) Cladogram.

**Figure 6 genes-14-01166-f006:**
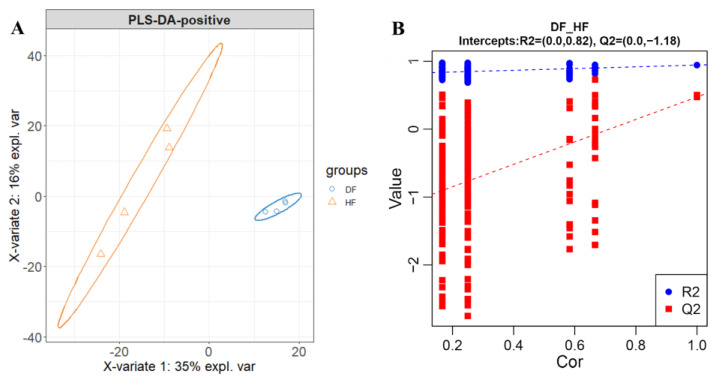
PLS-DA plot (**A**) and permutation test (**B**) derived from the fecal metabolite profiles of healthy and diarrheic piglets at positive ions mode. Differential analysis at negative ion mode refers to Appendix A.

**Figure 7 genes-14-01166-f007:**
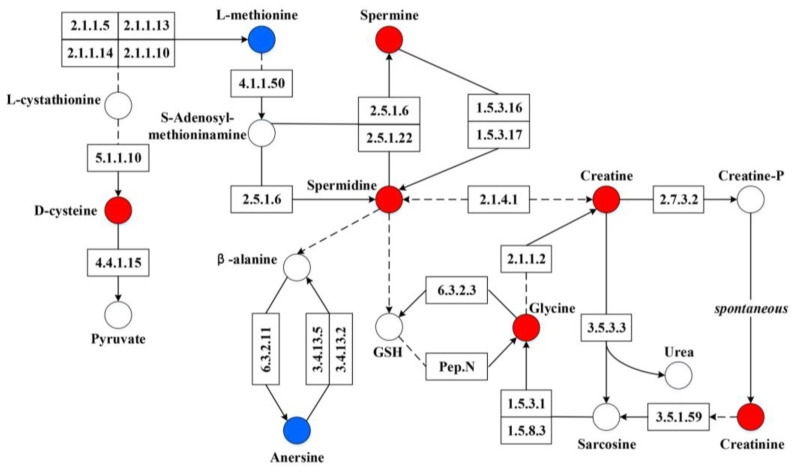
Sketch of arginine and proline metabolism, β-alanine metabolism, cysteine and methionine metabolism, glutathione metabolism, and glycine, serine, and threonine metabolism. Red circles indicate upregulated metabolites. Blue circles indicate downregulated metabolites. Dotted lines/arrows indicate indirect action. Solid lines/arrows indicate direct action.

**Figure 8 genes-14-01166-f008:**
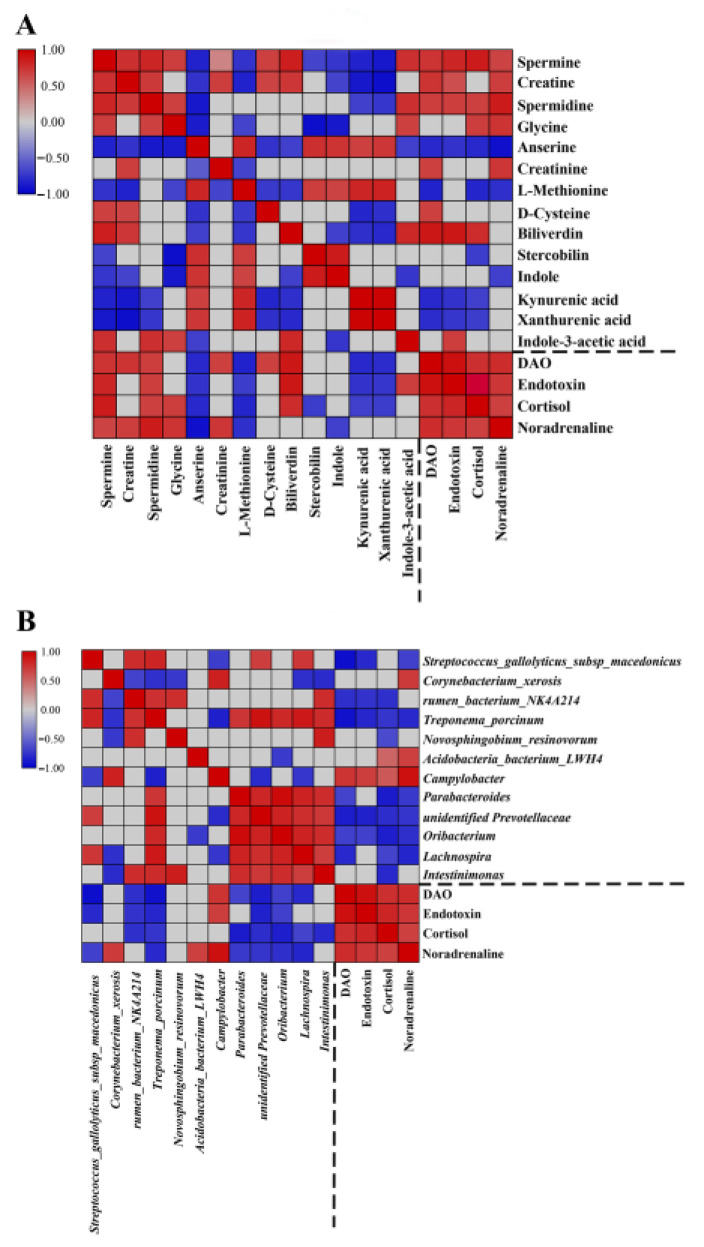
Heat map summarizing the correlation of differential gut microbiota, changed hematological indices, and altered fecal metabolites between the healthy and diarrheic groups. ((**A**) fecal metabolites vs. hematological indices (**B**) gut microbiota vs. hematological indices).

**Figure 9 genes-14-01166-f009:**
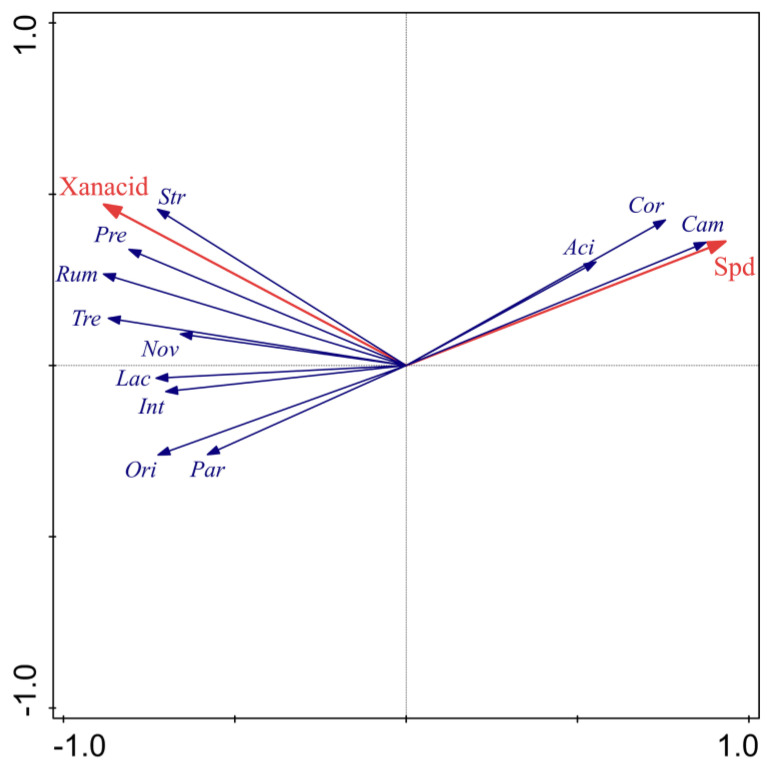
RDA analysis of altered metabolites on the community composition at genus and species level. Red arrows indicate altered metabolites with conditional effect of more than 10%. Blue arrows indicate differential gut microbiota at genus and species level. Spd: Spermidine, Xanacid: *Xanthurenic acid*, Cam: *Campylobacter*, Cor: *Corynebacterium xerosis*, Aci: *Acidobacteria bacterium* LWH4, Str: *S. macedonicus*, Pre: unidentified *Prevotellaceae*, Rum: *Rumen bacterium* NK4A214, Tre: *Treponema porcinum*, Nov: *N.resinovorum,* Lac: *Lachnospira*, Int: *Intestinimonas*, Ori: *Oribacterium*, Par: *Parabacteroides*.

## Data Availability

The 16S rRNA gene sequencing reads have been deposited in the Sequence Read Archive (SRA) database in the National Center for Biotechnology Information (NCBI) under the BioProject number PRJNA724903. The metabolomics sequencing data were submitted to the figshare database on the following link: https://figshare.com/articles/dataset/LC-MS_MS_data/12982259 (accessed on 7 January 2021).

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
