# Peer review of "Fecal Microbial Structure and Metabolic Profile in Post-Weaning Diarrheic Piglets"

_genes, 2023, doi:10.3390/genes14061166_

Round 1

Reviewer 1 Report

Comments and suggestions for authors

The manuscript titled „Fecal Microbial Structure and Metabolic Profile in Post-Wean
ing Diarrheic Piglets” by Zheng, Nie, Xu, Zhang, Xie, Xu, Zhang, Yueyun, Ding, Yin and Zhang, investigate the alteration of the fecal metabolic profile and the gut microbial structure in post-weaning piglets with diarrhea vs. health, the diarrea being a significant issue that can affects the entire pig production cycle.

The study has considerable scientific and practical relevance since it provides new insight on preventing or treating diarrhea in pigs as well as into the mechanisms that depend on the microbial population and how they relate to health and disease.

The references are relevant for the topic.

Specific comments:

Subsection 2.2. Please, rename the section since you described the research design and farm conditions in addition to the sample collection.

Subsection 2.3. Rename title „Changes of intestinal morphology in diarrheal and healthy piglets” It seems to me rather a title of Results than of Material and method

 Line 97. Please provide information about how long the trial was conducted. Were the piglets kept in individual cages? Why was there such a little number of animals? Statistically speaking, it was assured?

Line 206. Please give the whole term before using an abbreviation, such as redundancy analysis for i.g. (RDA) 

Line 112. Why ten samples? Please give more information here.

Subsection 2.2. Line 121 ... as detected using spectrophotometry based on the...please provide the authors name!

Results. In my opinion it makes the manuscript easier to comprehend and follow if the same structure is used throughout.

Line 271-272.  I sugest the character in figure 4 to be enlarged because it's difficult to read.

Discussion. Paragraph 369-371 ” In the present, we detected the effect of post-weaning diarrhea on the gut microbiota and fecal metabolomic profile was explored by utilizing 16S rRNA gene sequencing combined with LC-MS based metabolomics approach”, please rephrase.

Author Response

Reviewer's report:

Title: Fecal Microbial Structure and Metabolic Profile in Post-Weaning Diarrheic Piglets

Comments and suggestions for authors

Reviewer number: 1

The manuscript titled ,Fecal Microbial Structure and Metabolic Profile in Post-Wean

ing Diarrheic Piglets" by Zheng, Nie, Xu, Zhang, Xie, Xu, Zhang, Yueyun, Ding. Yin and Zhang,investigate the alteration of the fecal metabolic profile and the gut microbial structure in post-weaning piglets with diarrhea vs. health, the diarrea being a significant issue that can affects the entire pig production cycle.

The study has considerable scientific and practical relevance since it provides new insight onpreventing or treating diarrhea in pigs as well as into the mechanisms that depend on themicrobial population and how they relate to health and disease.

The references are relevant for the topic.

Specific comments:

Subsection 2.2. Please, rename the section since you described the research design and farm conditions in addition to the sample collection.

Response: Thanks for your suggestions. we have renamed the section title study design and animals.(please see line 96 on page 2)

Subsection 2.3. Rename title ,Changes of intestinal morphology in diarrheal and healthy piglets"It seems to me rather a title of Results than of Material and method

Response: Thanks for your suggestions. we have renamed the section title Intestinal morphology detection.(please see line 119 on page 3)

Line 97. Please provide information about how long the trial was conducted. Were the piglets kept in individual cages? Why was there such a little number of animals? Statistically speaking, it was assured?

Response: Thanks for your insightful comments. We have provided brief information about the trial period within article. It is presented by the yellow highlighting, please see page 3, line 102-109.

We totally agree with you that the number of animals is few. Unfortunately, we are truly sorry that there are no more suitable piglets to use. In order to reduce the influence of other factors on the intestinal microbiota, the animals here were siblings or half-siblings, with similar birth dates and parity, and were kept in same environment, which makes it difficult to obtain more suitable samples. To improve the reliability, we used reasonable statistical analysis methods. And we will carefully consider your suggestions in the future experimental design.

Thanks!

Line 206.Please give the whole term before using an abbreviation, such as redundancy analysis for i.g.(RDA)

Response: Thank you very much for your kindly remarks. Done.

Line 112.Why ten samples? Please give more information here.

Response: Thank you very much for your kindly reminder.We wrote it incorrectly by accident.

Subsection 2.2. Line 121 ... as detected using spectrophotometry based on the..please provide the authors name!

Response: Thank you very much for your kindly reminder.Done.

Results. In my opinion it makes the manuscript easier to comprehend and follow if the same structure is used throughout.

Response: Thanks a lot for your suggestion. We have carefully revised the results section according to your guidance.

Line 271-272. I suggest the character in figure 4 to be enlarged because it's dificult to read.

Response: Thank you very much for your kindly reminder.Done.

Discussion. Paragraph 369-371”In the present, we detected the effect of post-weaningdiarrhea on the gut microbiota and fecal metabolomic profile was explored by utilizing 16S rRNA gene sequencing combined with LC-MS based metabolomics approach", please rephrase.

Response: Thanks for your suggestion. We have rephrased the sentence and it is presented by the yellow highlighting, please see page 14.

Reviewer 2 Report

GENERAL COMMENTS

 In the study genes-2347193 entitled “Fecal Microbial Structure and Metabolic Profile in Post-Weaning Diarrheic Piglets.” the authors investigate the differences in the gut microbial structures and fecal metabolic profile from diarrheic and healthy weaned Wannan Black piglets, to identify the potential metabolic biomarkers of post-weaning diarrhea. The manuscript presents interesting new data, and that the topic of the manuscript is of significant interest and appropriate for the Journal. The manuscript is written in an acceptable English language. The presentation and the length of the manuscript are adequate as the description of the experimental plan. In particular, the title accurately reflects the major findings of the work; the keywords represent the article adequately and the abstract section well summarizes the background, methodology, results, and significance of the study; the introduction section is well written and supported by adequate bibliographic information, and it falls within the topic of the study, however, the purpose of the study could be clearly stated; material and methods section is well written and adequately and meticulously describes the methods applied in the study; results section is clear and the obtained findings were well explained. The discussion and the conclusion sections clear. Many paragraphs are too long, please rewrote these sections. However, some major changes I suggest throughout the text.

 AUTHORS COMMENTS

 This manuscript genes-2347193 the differences in the gut microbial structures and fecal metabolic profile from diarrheic and healthy weaned Wannan Black piglets, to identify the potential metabolic biomarkers of post-weaning diarrhea.

The title accurately reflects the major findings of the work.

The abstract clearly summarize the background, methodology, results, and significance of the study.

As authors referred in Introduction, identifying the potential metabolic biomarkers of post-weaning diarrhea will be beneficial to check the animal condition. Your findings are interesting; experimental aims might be relevant to the purpose of the journal in the present form. However, AA should improve some sentences. The introductory section is well written; however, the purpose of the study could be clearly stated.

The section of Materials and Methods is clear for the reader, but the authors should check this section and correct many punctuation errors.

Please rewrite discussion of the study.

Tables and Figures are generally good, and they well represent results obtained. Data in Tables were not duplicated in the text.

The reference list should be improved. Correct and uniform the references to the journal style.

Author Response

Zongjun-Yin, Ph.D.

Department of animal genetics and breeding

Anhui Agricultural University

Dear editors and reviewers:

We thank you for giving us an opportunity to revise our manuscript “Fecal Microbial Structure and Metabolic Profile in Post-Weaning Diarrheic Piglets

 [ID genes-2347193]”. We greatly appreciated you have made insightful comments and thoughtful guidance to our work which greatly improve our manuscript. Accordingly, we have carefully addressed all concerns pointed out by you. We detailed our responses to all comments point-by-point in the attached pages. For ease of reviewing, all significant changes in the revision were labelled in yellow.

We hope that our revision has satisfactorily addressed the comments and will be acceptable for publication in Genes.

Sincerely,

Zongjun-Yin, Ph.D.

Department of animal genetics and breeding

Anhui Agricultural University

Email: yinzongjun@ahau.edu.cn

Phone No.: 86-0551-65787303

Reviewer's report:

Title: Fecal Microbial Structure and Metabolic Profile in Post-Weaning Diarrheic Piglets

Comments and suggestions for authors

Reviewer number: 2

GENERAL COMMENTS

In the study genes-2347193 entitled "Fecal Microbial Structure and Metabolic Profile in Post-   Weaning Diarrheic Piglets." the authors investigate the differences in the gut microbial structures and fecal metabolic profile from diarrheic and healthy weaned Wannan Black piglets, to identify  the potential metabolic biomarkers of post-weaning diarrhea. The manuscript presents interesting new data, and that the topic of the manuscript is of significant interest and appropriate for the Journal. The manuscript is written in an acceptable English language. The presentation and the length of the manuscript are adequate as the description of the experimental plan. In particular, the title accurately reflects the major findings of the work; the keywords represent the article adequately and the abstract section well summarizes the background, methodology, results, and significance of the study; the introduction section is well written and supported by adequate bibliographic information, and it falls within the topic of the study, however, the purpose of the study could be clearly stated; material and methods section is well written and adequately and meticulously describes the methods applied in the study; results section is clear and the obtained findings were well explained. The discussion and the conclusion sections clear. Many paragraphs are too long, please rewrote these sections. However, some major changes I suggest throughout  the text.

AUTHORS COMMENTS

This manuscript genes-2347193 the differences in the gut microbial structures and fecal   metabolic profile from diarrheic and healthy weaned Wannan Black piglets, to identify the

potential metabolic biomarkers of post-weaning diarrhea.

The title accurately reflects the major findings of the work.

The abstract clearly summarize the background, methedology, results, and significance of the study.

As authors referred in Introduction, identifying the potential metabolic biomarkers of post-weaning diarrhea will be beneficial to check the animal condition. Your findings are interesting;

experimental aims might be relevant to the purpose of the journal in the present form. However,

AA should improve some sentences. The introductory section is well written; however, the purpose of the study could be clearly stated.

Response: Thanks a lot for your suggestion. We have added the purpose of the study in the introduction section according to your guidance.It is presented by the yellow highlighting, please see lines 80-86 on page 2.

The section of Materials and Methods is clear for the reader, but the authors should check this section and correct many punctuation errors.

Response: Thank you very much for your kindly reminder. We have carefully revised the punctuation errors in the article.

Please rewrite discussion of the study. 

Tables and Figures are generally good, and they well represent results obtained. Data in Tables were not duplicated in the text.

The reference list should be improved. Correct and uniform the references to the journal style.

Response: Thank you for your suggestion. Done.

Reviewer 3 Report

The manuscript titled Fecal microbial structure and metabolic profile in post-weaning diarrheic piglets by Zheng et al investigated the microbiome and metabolomics of 4 diarrheic pigs compared to 4 control pigs. Since by their own admission, diarrhea can have multiple etiologies, the scope of this work is very limited. There is no data from other farms to indicate that this work is translatable to another swine operation. Furthermore, we really do not know if the change in bacteria is the cause or consequence of diarrhea, or if that microbiome happens to be that of those animals in absence of diarrhea. I feel this limitation of the research must be acknowledged. The animal numbers are very low.

Other considerations

Line 51. Suggest “It has been suggested…” rather than “several researches suggested”

Line 60 – 61. Unclear what the difference of “abiotic or abiotic interference”

Line 82. States 5 diarrheic pigs contradicts line 95 stating 4 diarrheic pigs.

Line 100. “was not” rather than “didn’t be”

Line 102. “breast fed” suggests human reference. Prefer “suckled”

Line 124. Add author and year for citation [21].

Line 160. Delete “And”

Line 173. Clarify what is meant by “appropriate supernatant”

Line 263. “did not differ” rather than “insignificant differences”

There are a few errors, but for the most part it is good.

Author Response

Zongjun-Yin, Ph.D.

Department of animal genetics and breeding

Anhui Agricultural University

Dear editors and reviewers:

We thank you for giving us an opportunity to revise our manuscript “Fecal Microbial Structure and Metabolic Profile in Post-Weaning Diarrheic Piglets

 [ID genes-2347193]”. We greatly appreciated you have made insightful comments and thoughtful guidance to our work which greatly improve our manuscript. Accordingly, we have carefully addressed all concerns pointed out by you. We detailed our responses to all comments point-by-point in the attached pages. For ease of reviewing, all significant changes in the revision were labelled in yellow.

We hope that our revision has satisfactorily addressed the comments and will be acceptable for publication in Genes.

Sincerely,

Zongjun-Yin, Ph.D.

Department of animal genetics and breeding

Anhui Agricultural University

Email: yinzongjun@ahau.edu.cn

Phone No.: 86-0551-65787303

Reviewer's report:

Title: Fecal Microbial Structure and Metabolic Profile in Post-Weaning Diarrheic Piglets

Comments and suggestions for authors

Reviewer number: 3

The manuscript titled Fecal microbial structure and metabolic profile in post-weaning diarrheic

piglets by Zheng et al investigated the microbiome and metabolomics of 4 diarrheic pigs

compared to 4 control pigs. Since by their own admission, diarrhea can have multiple etiologies, the scope of this work is very limited. There is no data from other farms to indicate that this work is translatable to another swine operation. Furthermore, we really do not know if the change in

bacteria is the cause or consequence of diarrhea, or if that microbiome happens to be that of

those animals in absence of diarrhea. I feel this limitation of the research must be acknowledged. The animal numbers are very low.

Response: Thank you very much for your insightful comments. We totally agree with you that the number of animals in this study is few. It was our thoughtlessness. Unfortunately, we are truly sorry that there are no more suitable piglets to use. In order to reduce the influence of other factors on the intestinal microbiota, the animals here were siblings or half-siblings, with similar birth dates and parity, and were kept in same environment, which makes it difficult to obtain more suitable samples. To improve the reliability, we used reasonable statistical analysis methods. And we will carefully consider your suggestions in the future research.

Thanks again!

Other considerations

Line 51.Suggest "It has been suggested. … ." rather than "several researches suggested"

Response: Thank you very much for your kindly suggestion, done.

Line 60-61.Unclear what the difference of"abiotic or abiotic interference"

Response: Thank you very much for your carefully suggestion, we are so sorry that the author mean different types of interference, abiotic or other factor interference. We have revised this mistake in the article. Please see line 62 on page 2.

Line 82.States 5 diarrheic pigs contradicts line 95 stating 4 diarrheic pigs.

Response: Thank you very much for your kindly reminder. We wrote it incorrectly by accident. We have removed the error information in the revised article.

Line 100."was not" rather than "didn't be"

Response: Thank you very much for your kindly reminder. Done. Please see line 103 on page 3.

Line 102."breast fed" suggests human reference. Prefer"suckled"

Response: Thank you very much for your kindly reminder. Done. Please see line 104 on page 3.

Line 124.Add author and year for citation [21].

Response: Thanks for your kindly reminder. Done. Please see line 131 on page 3.

Line  160.Delete"And"

Response: Thanks for your kindly reminder. Done. Please see line 168 on page 4.

Line 173.Clarify what is meant by "appropriate supernatant'

Response: Thanks for your kindly reminder. We didn't express it accurately enough, it means enough to do experiments.

Line 263."did not differ? rather than "insignificant differences"

Response: Thanks for your kindly reminder. Done.

Round 2

Reviewer 2 Report

The manuscript is now suitable for publication

Author Response

 Dear reviewer,

Thank you very much for your kindly comments. 

Kindly regards,

Xianrui Zheng

Reviewer 3 Report

The limitations of this study are not adequately addressed. In the conclusion section results need to be tempered with the limited scope of this data. The microbiome is highly variable and subject to environment. It should be written in the results that this has a limited scope since all pigs (numbers are small) were from the same farm and were in fact siblings or half siblings. These results have the most significance to this farm. This limitation has to be addressed in the discussion and conclusion.

Other:

L 83. "...mechanisms [are] still not clear..."

L83-85. Run on sentence. Suggest revision by breaking the new sentence at "There are no reports..."

Line 454. "profiles" is misspelled.

It is acceptable.

Author Response

Reviewer  3

The limitations of this study are not adequately addressed. In the conclusion section results need to be tempered with the limited scope of this data. The microbiome is highly variable and subject to environment. It should be written in the results that this has a limited scope since all pigs (numbers are small) were from the same farm and were in fact siblings or half siblings. These results have the most significance to this farm. This limitation has to be addressed in the discussion and conclusion.

Response: Thank you very much for your kindly suggestions. We have added the limitations according to your guidance, please see L406-408 on page 14 in the discussion section and L494-496 on page 16 in the conclusion section.

Other:

L 83. "...mechanisms [are] still not clear..."

Response: Thank you very much for your kindly reminder. Done.

L83-85. Run on sentence. Suggest revision by breaking the new sentence at "There are no reports..."

Response: Thank you very much for your kindly suggestion. We have revised according your guidance, please see L83-85 on page 2.

Line 454. "profiles" is misspelled.

Response: Thanks for your kindly reminder. We have revised within the article.